# Photodynamic Therapy Used to Treat an HIV Patient with an Efflorescence of Viral Warts after HAART Therapy

**DOI:** 10.3390/diagnostics13061098

**Published:** 2023-03-14

**Authors:** Francesca Ambrogio, Alexandre Raphael Meduri, Giovanni Infante, Melita Anna Poli, Paolo Romita, Domenico Bonamonte, Raffaele Filotico, Giuseppe Ingravallo, Gerardo Cazzato, Carmelo Laface, Aurora De Marco, Caterina Foti

**Affiliations:** 1Section of Dermatology and Venereology, Department of Precision and Regenerative Medicine and Ionian Area (DiMePRe-J), University of Bari “Aldo Moro”, 70124 Bari, Italy; 2Infectious Diseases, ASL BAT, 76011 Bisceglie, Italy; 3Section of Molecular Pathology, Department of Precision and Regenerative Medicine and Ionian Area (DiMePRe-J), University of Bari “Aldo Moro”, 70124 Bari, Italy; 4IRCCS Istituto Tumori “Giovanni Paolo II”, 70124 Bari, Italy

**Keywords:** HIV, photodynamic therapy, HAART, wart

## Abstract

Healing from viral warts lesions can be hard to achieve in immunocompromised subjects like HIV-positive patients. The therapeutic target in immunocompetent subjects can be reached using different methods, including topical ointments, cryotherapy, laser therapy, imiquimod, and photodynamic therapy (PDT). We present a case of a male HIV-positive patient who came to the Dermatology department with multifocal wart lesions on his face, auricular, and retro-auricular areas after treatment with highly active antiretroviral therapy (HAART). In our case, surprisingly, only one session of PDT proved to induce complete regression of lesions which, despite their thickness, had a much more robust response to treatment than we could have possibly expected. After a brief review of the literature, it is possible to state that PDT revealed itself to be a valid option in immunocompromised patients who have a major risk of relapse.

## Figures

**Figure 1 diagnostics-13-01098-f001:**
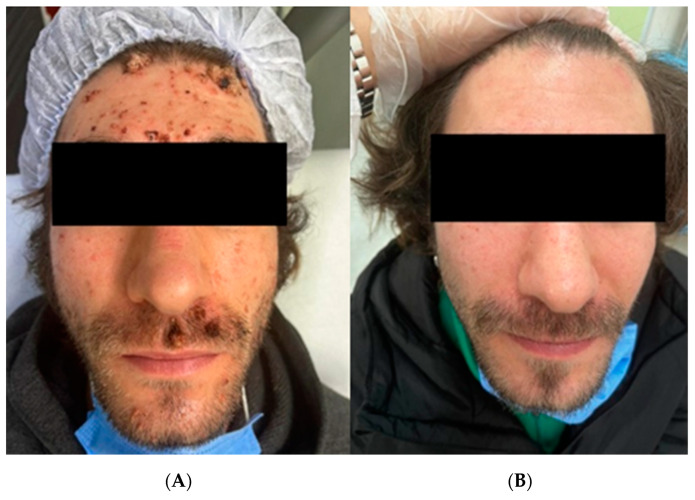
(**A**) Patient before the treatment. (**B**) Patient after one session of PDT. In early November 2022, a thirty-one-year-old male patient came to the Dermatology department with multifocal wart lesions on his face (**A**), auricular (Figure 2A), and retro-auricular areas (Figure 2B). The anamnesis disclosed that in February 2022, the patient had started an antibiotic therapy with sulfamethoxazole and trimethoprim once a day, as well as HAART therapy with darunavir, cobicistat, emtricitabine, and tenofovir alafenamide after a blood sample tested positive for HIV infection on 21/02/2022 with a CD4 T-helper lymphocyte count of 0.76% (NV: 35–55%) 3/µL, a CD8 T lymphocyte count of 32.23% (VN: 15–35%) 116/μL, and an HIV RNA of 680,000 cp/mL. Echography did not show any abnormality except a reactive lymphadenopathy. During his first visit to our department, the patient’s CD4 T-helper lymphocytes count had already increased up to 13.27% (NV: 35–55%) 128/µL, the CD8 T-suppressor lymphocyte count was 49% (normal value 15–35%) 472/μL, and he had a null viral load. After assessing the psychological impact of the lesions on patient life through the Dermatology Life Quality Index (DLQI) and finding a score of 21, we performed a skin biopsy with subsequent histological examination and molecular typing of HPV strains trough PCR. The results confirmed the diagnostic of viral warts was positive for HPV 11 and 16. Healing from viral wart lesions can be hard to achieve in immunocompromised subjects such as HIV-positive patients [1]. In immunocompetent subjects, warts can heal without treatment during the first year in about half of the cases [2]; the rate of clearance is influenced by factors such as viral type, the host’s immune status, and the duration of warts [3]. The therapeutic target can be achieved using different methods, including topical ointments, cryotherapy, laser therapy, imiquimod, and photodynamic therapy (PDT) [4]. All techniques used in wart management are associated with varying degrees of recurrence rates [5] and have different indications and different degrees of patient tolerance and acceptance due to their painful or distressing application. Treatment of warts can be less efficient in patients with HIV, with a longer duration of therapy or with relapse of the disease due to the insufficient ability of the body to contrast viral progression [6]. In addition, warts are socially perceived as unpleasant, and they can represent stressful issues for the patient who seeks definitive results in the shortest possible time. The formation of viral warts is one of the consequences of infection from human papillomavirus (HPV), a worldwide-spread virus that can be classified either as high-risk, such as strains 16, 18, 31, 33, 35, 39, 45, 51, 52, 56, 58, 59, and 66, or low-risk, such as strains 6, 11, 40, 42, 43, 44, 54, 61, 70, 72, 81, and 89 [7]. HPV strains 6 and 11 are the most frequently found in genital warts, also known as condyloma acuminate. On the other hand, HPV strains 1, 2, 4, and 7 are more commonly detected on the face with the polymerase chain reaction (PCR) technique (1). Another consequence of HPV infection, especially in immunocompromised patients, is the transformation of the common wart into a malignant, neoplastic lesion which can be life-threatening if not addressed promptly, and at times vigorously, with adequate treatment. PDT is a technique that exploits a substance called photosensitizer and a light source [6,8,9]. The photosensitizer is a molecule that is mainly absorbed by cells with increased metabolic activity and will cause oxidative damage in infected cells once the light ray irradiates the target area of the skin. It must be applied on the lesions for a certain amount of time before irradiating the area to ensure its efficacy. Along with the photosensitizer, the time of irradiation from the light source must be accurately set to deliver a specific load of energy. During treatment, a burning sensation can be perceived by the patient; therefore, a break or a spray of water may be required to ease the patient’s pain [8]. This technique is already used to treat superficial non-melanoma skin cancer, such as basal cell carcinoma or actinic keratosis [6]. Highly active antiretroviral therapy (HAART) is used in HIV patients to contrast viral replication and reverse CD4 lymphocyte depletion [10]. Additionally, it is well-established knowledge that an adequate CD4 lymphocyte count is essential to skin immunity. A stunted immune response can lead to a higher number of skin diseases. HPV infection in HIV-positive patients tends to manifest with multiple warts. It is more resistant to traditional therapy, thus leading to longer periods of treatment and a higher number of recurrences.

**Figure 2 diagnostics-13-01098-f002:**
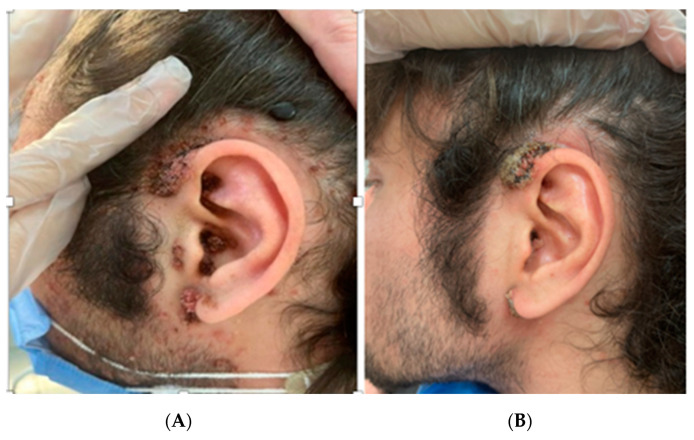
(**A**) Left auricular area before treatment. (**B**) Left auricular area after two sessions of PDT. Only a single session was needed to treat the facial warts (Figure 1B). Interestingly, little warts located on the untreated auricular area and on the scalp also disappeared without MAL gel application (**B**). The retro-auricular warts disappeared after only a single PDT session (Figure 3B). There was no recurrence of the disease nor any kind of skin dyschromia after a three-month follow-up. We then asked the patient to fill in the DLQI form again and obtained a score of 0. Upon consent of the patient, we performed C-PDT using the photosensitizing agent methyl aminolevulinate (METVIX^®^ cream, Galderma Medical Solutions), left the affected area under occlusion for three hours, and then illuminated the affected area with a red 630 nm light-emitting diode (LED) lamp (Aktilite CL128^®^, Galderma Uppsala, Sweden) set at 37 J/cm^2^ for 8:30 min. The patient felt a very mild burning sensation that was relieved with a spray of water. Because of the high number of lesions in different areas, and in order to avoid too much discomfort, we decided to initially treat only lesions on the face.

**Figure 3 diagnostics-13-01098-f003:**
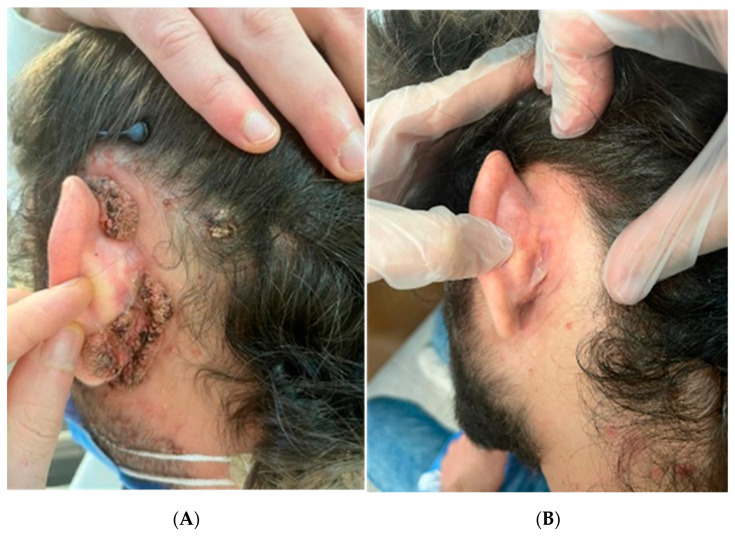
(**A**) Left retro-auricular area before PDT. (**B**) Untreated left retro-auricular area. Clinical indications for PDT are increasingly growing within the literature, and one of these indications is to treat verrucae [9,11,12,13]. It is clear that the regression rate of this sort of lesion is very variable. A lot of research is currently being carried out in the field of photosensitizers to bring greater selectivity through biotechnology-based breeding techniques, such as targeted genome editing methods [14]. In fact, even in the case of common hand and foot warts, clearance rates have been reported to be extremely variable, with values between 56% and 100% after repetitive treatments (up to six sessions) of PDT [6]. In our patient, cellular mediated immunity was impaired, PCR showed the presence of high-risk HPV, and lesions appeared quickly only 4 weeks prior to consultation. These are not good indicators for self-clearance in a short amount of time [2]. Therefore, we decided to discuss the best treatment options with our patient. First-line treatment in the literature for facial warts consists of topical salicylic acid and cryotherapy [4], but after consultation with the patient and owing to the important risk of dyschromia, scars, and protracted treatment, we agreed to first try PDT. In our case, surprisingly, only one session of PDT proved to induce complete regression of lesions which, despite their thickness, had a much more robust response to treatment than we could have possibly expected. Thickness of lesions and immune impairment of treated patients are two of the major factors of ineffective response to PDT treatments in clinical studies; additionally, the site of the lesion, the machine’s adjustments, and the time of incubation of the photosensitizer seem to be of influence [15]. In our case, PDT did not leave any noticeable mark on the face at a three-month follow-up, and moreover, no relapse of lesions has been reported. Compared to immunocompetent patients, individuals who present some form of immune impairment, like human immunodeficiency virus (HIV)-positive subjects, tend to have a higher number of viral lesions and recurrences caused by HPV. They also typically have a poor reaction to conventional treatments due to impaired cellular immunity [16]. In the course of our bibliographic research, we found very few studies about the use of PDT on immunocompromised patients [15], and none of them focused on treatment of eruptive facial, auricular, nor retro-auricular warts in HIV patients during HAART therapy. Occurrence of these eruptive warts may be explained as a consequence of immune reconstitution inflammatory syndrome (IRIS). Interestingly, another instance of an IRIS-like syndrome has been reported in the literature, although in this case it occurred concurrently with the HPV vaccine [17]. Since the immune reconstitution phase of the patient following HAART was still ongoing, and without a fully regained count of CD4+ lymphocytes, we hypothesised that the number of antigens exposed upon application of PDT therapy was high enough to compensate for the deficiency of CD4+ lymphocytes and to trigger stimulation of CD8+ cells. The favourable response may be due to a larger antigen expression from HPV-infected cells after PDT, thus stimulating an effective CD8+ lymphocyte response. Indeed, in our case, even an untreated auricular and retro-auricular wart disappeared, probably thanks to the immune CD8+ mediated response. Our patient did not relapse after three months and did not show any hyperchromatic or hypomelanotic features on the skin, leading us to believe that the protocol of PDT was optimal for wart lesions on the head, but other studies recruiting a larger number of patients are needed to confirm our results. PDT still needs to be standardised with various parameters in order to be adapted to the daily practice of physicians for personalised medicine. However, our report shows how impressive results can be obtained, especially in this case, where extension of the lesions would have made other kinds of therapy very difficult to apply due to their effect on the patient’s skin. PDT could be a valid option in immunocompromised patients who have a major risk of relapse, even if more studies are necessary to corroborate our results.

## Data Availability

Not applicable.

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
