# Peer review of "Photodynamic Therapy Used to Treat an HIV Patient with an Efflorescence of Viral Warts after HAART Therapy"

_diagnostics, 2023, doi:10.3390/diagnostics13061098_

Round 1
Reviewer 1 Report
The work by Gerardo Cazzato and co-workers entitled "Photodynamic therapy used to treat an HIV patient with an efflorescence of viral warts after HAART therapy"describes the Healing from viral warts lesions in immunocompromised subjects like HIV-positive patients. The authors presented a case of a male HIV-positive patient with multifocal wart lesions on his face, auricular and retro-auricular area after treatment with highly active antiretroviral therapy (HAART) and only one session of PDT proved to induce complete regression of lesions which, despite their thickness, had a much more robust response to treatment. However, there are major comments to be addressed before its publication in the present journal.
1) Though this is a preliminary study, with 3 months regression tested, high chances of regression on HPV warts are observed within 6 months and 1 year. Justify
2) What were the negative effects observed in patients with this therapy?
3) How were the CD4 and CD8 levels measured? Provide results.
4) Did any scaling factors were considered to measure the warts? Results provided were just quantitative but not qualitative
5) Also, the population size is too small (only 1) which is not convincing
6) Is the patient given with any other medications? If so mention them.
Author Response
Reviewer 1
We would like to thank the Reviewer N1 for his constructive criticisms.
The first part of the objections raised concerned the need to revise our methodological approach in order to improve this clinical case report. We thus performed extensive changes in our original submitted text following the reviewer’s indications and corrected the language issues.
We have changed our text accordingly to the reviewer’s suggestions and questions. We include the following detailed explanation of our arguments
1) Though this is a preliminary study, with 3 months regression tested, high chances of regression on HPV warts are observed within 6 months and 1 year. Justify
The reviewer his right, many warts regress spontaneously, especially in children, but also in adults. The rate of clearance is influenced by factors such as viral type, host immune status and duration of warts.
https://www.ncbi.nlm.nih.gov/pmc/articles/PMC1764803/
However, in this clinical case report,
-The patient presented an impairment of the cellular mediated immunity
-A PCR showed the presence of multiple HPV strains ,including a high risk one
- The lesions had occurred only for weeks before, thus an indeterminate amount of time might be required for an eventual spontaneous resolution.
The patient came in our department in distress because of the unesthetic lesions of his head, , we found it appropriate to treat the patient quickly to relieve him of his
2) What were the negative effects observed in patients with this therapy?
à The patient tolerated the therapy very well, experiencing only a mild burning sensation promptly resolved with a spray of water.
3) How were the CD4 and CD8 levels measured? Provide results.
At the moment of the HIV diagnosis:
- Anti HIV1-2 on Chemiluminescence: positive
- CD4 T Helper lymphocyte 0.76% (VN: 35-55%) 3/L
- CD8 T Suppressor lymphocyte 32.23% (VN : 15-35%) 116/L
- HIV-RNA= 680 000 cp/mL (by PCR)
Results of the patient’s last blood sample analysis before consulting at our department:
- CD4 T Helper lymphocytes = 13,27% (normal values 35-55%), 128/L
- CD8 T Suppressor lymphocytes = 49% (normal values 15-35%), 472/L
- HIV-RNA null (by PCR)
4) Did any scaling factors were considered to measure the warts? Results provided were just quantitative but not qualitative.
à We used DLQI to measure the impact of his dermatological condition on the well been of the patient. We found an initial score of 21 before therapy and of 0 after PDT.
5) Also, the population size is too small (only 1) which is not convincing
à Our clinical single case report was meant to show the surprising prompt resolution of defigurating wart lesions on the visage after treatment with photodynamic therapy in a HIV+ patient undergoing immune reconstitution, it is not meant to replace observational studies or other more evidenced-base papers on the subject.
There is very little bibliographic material about photodynamic therapy in IRIS patient and we hope our report may offer some clues on the treatment of similar patients with a very well tolerated technique.
6) Is the patient given with any other medications? If so mention them.
à from february 2022 to now, the patient has been subjected to:
-Daily antibiotic therapy with Sulfamethoxazole and trimethoprim once a day
-Daily HAART therapy with Darunavir, Cobicistat, Emtricitabine and Tenofovir alafenamide.
Reviewer 2 Report
The type article "Interesting image" presented by Ambrogio F. et al. entitled Photodynamic therapy used to treat an HIV patient with an efflorescence of viral warts after HAART therapy fits the requirements to be published in this type of article in the Diagnostics journal. However, some aspects must be addressed and included in the report.
1. Relevant referent papers related PDT and PDT treatment for the treatment of viral warts must be included to support or contrasted findings. As authors mentioned, they included a brief review of literature to discuss the results. Please see the following references: doi: 10.1159/000323215, DOI: 10.1111/j.1346-8138.2009.00694.x, DOI: 10.1016/j.phyplu.2021.100044, DOI:10.1016/j.pdpdt.2022.102913, DOI: 10.1007/s10103-019-02823-3, DOI: 10.1080/14764172.2020.1785626.
2. Please define the type of photosensitizer used. Only M-ALA is mentioned.
3. Was the PS cream prepared from a pure drug? A commercial cream was used? Please detail the application protocol.
4. Please describe the irradiation apparatus with more specifications (eg. power density). Is it a commercial or homemade machine? Illustrations of PDT procedures could be included.
Author Response
Reviewer 2
The type article "Interesting image" presented by Ambrogio F. et al. entitled Photodynamic therapy used to treat an HIV patient with an efflorescence of viral warts after HAART therapy fits the requirements to be published in this type of article in the Diagnostics journal. However, some aspects must be addressed and included in the report.
1) Relevant referent papers related PDT and PDT treatment for the treatment of viral warts must be included to support or contrasted findings. As authors mentioned, they included a brief review of literature to discuss the results. Please see the following references: doi: 10.1159/000323215, DOI: 10.1111/j.1346-8138.2009.00694.x, DOI: 10.1016/j.phyplu.2021.100044, DOI:10.1016/j.pdpdt.2022.102913, DOI: 10.1007/s10103-019-02823-3, DOI: 10.1080/14764172.2020.1785626.
à We are honoured to receive these favourable comments about our case report. As suggested, in this new review we sought to explain our case report better by adding the bibliographic notes mentioned.
2) Please define the type of photosensitizer used. Only M-ALA is mentioned.
3) Was the PS cream prepared from a pure drug? A commercial cream was used? Please detail the application protocol.
4) Please describe the irradiation apparatus with more specifications (eg. power density). Is it a commercial or homemade machine? Illustrations of PDT procedures could be included.
à We performed C-PDT using the photosensitizing agent methyl aminolevulinate (METVIX® cream, Galderma Medical Solutions), left under occlusion for three hours, then illuminated with a red 630 nm light-emitting diode (LED) lamp (Aktilite CL128®, Galderma) set at 37 J/cm2 for 8:30 minutes. Red light irradiation with the Aktilite CL 128 and Metvix (Galderma SA) as a photosensitizing molecule is a conventional protocol approved and widely used in Europe for PDT treatment of actinic keratosis but much used in other diseases such as viral warts.
Reviewer 3 Report
The authors presented a case study of using PDT for wart lesion treatment. There are three pictures shown. as before and after comparison, and that it is. There is no sort of data or analysis presented. It is also hard to see any data to improve diagnostics. It is a wart lesion. Maybe it is better suited for another journal.
Author Response
Reviewer 3
The authors presented a case study of using PDT for wart lesion treatment. There are three pictures shown. as before and after comparison, and that it is. There is no sort of data or analysis presented. It is also hard to see any data to improve diagnostics. It is a wart lesion. Maybe it is better suited for another journal.
à We believe this is an interesting case report to present in the literature not only for the therapeutic method used that has obtained excellent results in a session without discomfort but to show the explosive reaction post HAART that has manifested itself. We tried to diagnose these eruptive lesions in the best possible way by histological examination and PCR of the lesions to provide the greatest possible knowledge of the reaction. We want to describe in the literature this case to increase the case history of this possible dermatological event after HAART. After the review we have hopefully made some useful changes and thank you for the incentives to improve our case reports
Round 2
Reviewer 2 Report
The authors have answered my comments and included the necessary changes. I have no more suggestions.